# A Novel L-Gate InGaAs/GaAsSb TFET with Improved Performance and Suppressed Ambipolar Effect

**DOI:** 10.3390/mi13091474

**Published:** 2022-09-05

**Authors:** Boyang Ma, Shupeng Chen, Shulong Wang, Tao Han, Hao Zhang, Chenyu Yin, Yaolin Chen, Hongxia Liu

**Affiliations:** Key Laboratory of Wide Band-Gap Semiconductor Materials and Devices of Education, The School of Microelectronics, Xidian University, Xi’an 710071, China

**Keywords:** band-to-band tunneling, InGaAs/GaAsSb, L-shaped gate, analog/RF performance

## Abstract

A heterojunction tunneling field effect transistor with an L-shaped gate (HJ-LTFET), which is very applicable to operate at low voltage, is proposed and studied by TCAD tools in this paper. InGaAs/GaAsSb heterojunction is applied in HJ-LTFET to enhance the ON-state current (I_ON_). Owing to the quasi-broken gap energy band alignment of InGaAs/GaAsSb heterojunction, height and thickness of tunneling barrier are greatly reduced. However, the OFF-state leakage current (I_OFF_) also increases significantly due to the reduced barrier height and thickness and results in an obvious source-to-drain tunneling (SDT). In order to solve this problem, an HfO2 barrier layer is inserted between source and drain. Result shows that the insertion layer can greatly suppress the horizontal tunneling leakage appears at the source and drain interface. Other optimization studies such as work function modulation, doping concentration optimization, scaling capability, and analog/RF performance analysis are carried out, too. Finally, the HJ-LTFET with a large I_ON_ of 213 μA/μm, a steep average SS of 8.9 mV/dec, and a suppressed I_OFF_ of 10^−12^ μA/μm can be obtained. Not only that, but the fT and GBP reached the maximum values of 68.3 GHz and 7.3 GHz under the condition of Vd = 0.5 V, respectively.

## 1. Introduction

The performance and density of integrated circuits have been significantly enhanced by scaling down of the MOSFET in the past several decades [1]. However, the conventional MOSFETs has a minimum subthreshold swing limited at 60 mV/dec. On the premise of maintaining device performance, scaling of the supply voltage will unavoidably be accompanied with the increasing I_OFF_, which will finally result in increased static power consumption. Therefore, researchers have to find other ways to solve the contradiction between static power consumption and performance [2,3,4]. Due to their steeper switching behavior than MOSFET, the tunneling field effect transistors (TFET), based on the band-to-band tunneling (BTBT) operation mechanism, have been considered as a potential substitute for ultra-low power applications [5,6,7,8,9].

However, low ON-state current, large leakage current, and poor analog performance are three deficiencies that conventional TFETs need to confront [10,11,12,13]. These need to further optimize the conventional TFET for low-power and high-frequency application. In recent years, many research groups have done valuable work to improve the performance of TFET. Many novel TFETs with various structures have been proposed, such as L-TFET [14,15,16], U-TFET, and TGTFET [17,18,19]. Due to larger tunnel junction area and stronger gate-controlled capability, higher ON-state current and lower subthreshold swing than conventional planar TFET can be obtained. In spite of these advantages, the low ON-state current is still one of the greatest challenges of the TFET design. TFET still cannot meet the demand in commercial applications. To further improve the ON-state current, multi-material heterojunction engineering is applied to improve the band-to-band tunneling rate [20]. Compared with conventional homojunction TFET, research shows that HJTFETs have superior performances, such as Si/SiGe heterojunction [21,22,23,24], SiGeSn/GeSn heterojunction [25], and III–V heterojunction [26,27,28,29,30,31,32]. By selecting appropriate materials to form heterojunction, on the one hand, the height and thickness of the tunneling barrier can be effectively reduced. On the other hand, smaller carrier mass and higher carrier mobility can be achieved than with Si-based TFET. In this way, the tunneling rate can be further improved, and the larger ON-state current can be obtained. However, for HJTFETs, large OFF current, poor analog performance, and bipolar effect still need to be improved.

In this work, a novel InGaAs/GaAsSb HJ-LTFET with a remarkably good switching performance and suppressed ambipolar effect is proposed. Benefiting from the quasi-broken gap energy band alignment of InGaAs/GaAsSb heterojunction and high tunneling efficiency of the L-shaped gate structure, HJ-LTFET obtained maximum I_ON_ of 213 μA/μm. Moreover, the insertion of barrier layer between the source and drain obviously suppressed the OFF-state source-to-drain tunneling on horizontal direction. On the other hand, the application of multiple-gate work functions by electrode work function modulation technics [33] obviously suppressed the OFF-state source-to-drain tunneling in the diagonal direction, which can help to achieve the I_OFF_ of 10^−12^ μA/μm.

## 2. Structure and Mechanism of the HJ-LTFET

Figure 1a shows the structure of the proposed InGaAs/GaAsSb HJ-LTFET. The tunneling junction consists of a p+ GaAs_0.9_Sb_0.1_ source with doping concentration of 2 × 10^19^ cm^−3^ and a n^+^ In_0.9_Ga_0.1_As channel layer with doping concentration of 1 × 10^18^ cm^−3^. To maximize the effective area of line tunneling junction, the source region and channel layer are overlapped as much as possible by structural design. The defined gate length (*L*_g_) and gate height (*H*_g_) are equal to 50 nm, and the thickness of channel (*T*_C_) is 10 nm. In order to suppress the source-to-drain tunneling (SDT) leakage current effectively, a 5 nm thickness HfO2 barrier layer is inserted to suppress the OFF-state tunneling between the source and the drain on the horizontal direction. On the other hand, the metal work function is modified to further reduce the OFF-state tunneling in other directions, such as vertical and diagonal directions. As shown in Figure 1, the metal work function of the dark purple region on the left is 4.75 eV, and that of the light purple region on the right is 4.85 eV. Furthermore, an HfO2 layer with 2 nm thickness is selected as the gate dielectric, which remarkably improves the gate control ability while maintaining enough gate dielectric thickness [34].

Figure 2 shows the ON-state and OFF-state energy band diagram of HJ-LTFET. The cutline positions are shown by the red dotted lines. In the ON-state, as shown in Figure 2a, electrons are tunneling from p+ GaAsSb source to n+ InGaAs channel. with the gate voltage increasing, the energy bands near the heterojunction will bend downward to meet the BTBT condition. After tunneling, the electrons will transfer from channel to drain and finally be collected by the drain electrode. In the OFF-state, as shown in Figure 2b, there no BTBT occurs, and no tunneling current is observed in HJ-LTFET.

Correspondingly, the electron and hole current density distribution also show the current switching mechanism of HJ-LTFET, as shown in Figure 3. It is not difficult to see from Figure 3a,b that a small amount of free electrons flows from the channel layer to the drain electrode, forming the *I*_OFF_. The OFF-state holes have no contribution to the I_OFF_ because the existence of the HfO2 barrier layer blocks the hole conductive path between the source and drain electrode. This indicates that in the OFF-state, there is no electron tunneling from the source region to the channel. Additionally, the presence of the space charge region results in significant resistance close to the p-n junction. However, in the ON-state, the electron current density and hole current density are extraordinarily high, as shown in Figure 3c,d. These electrons and holes mainly tunnel from heterojunction.

The transfer characteristic curve simulated by Silvaco TCAD is shown in Figure 1b. For the InGaAs/GaAsSb material used in the simulation, we obtained the fitting parameters in [35]. In order to accurately calculate the tunneling process under reverse bias condition, the hurkx BTBT model was used in this work, and models of Fermi–Dirac distribution, SRH recombination, non-local BTBT and field-dependent mobility were also used along with a band gap narrowing model. Considering both the high-doped source region and the low-doped channel, the BTBT parameters used to calculate the tunneling current are *A*_BTBT=_ 1.3 × 10^20^ (cm^−3^ s^−1^), and *B*_BTBT=_ 5.7 × 10^6^ (V cm^−1^), respectively [36].

## 3. Discussion of Simulation Results

### 3.1. Effect of Doping Concentration in Source, Channel, and Drain on Device Performance

Figure 4 shows the change of the transfer characteristic, ON-state current (I_ON_), and switching current ratio (I_ON_/I_OFF_) of HJ-LTFET with different source doping concentrations (N_S_). The subthreshold region shifts to the left with the increase of NS, as shown in Figure 4a. This is because the higher the N_S_ value, the easier the formation of the BTBT channel, and the smaller the V_GS_ value required to turn ON the BTBT process. When NS is more than 2.0 × 10^19^ cm^−3^, an OFF-state BTBT phenomenon can be found near the source/channel interface in Figure 4b, which can cause a large I_OFF_ and a small I_ON_/I_OFF_. When N_S_ is lower than 2.0 × 10^19^ cm^−3^, the BTBT leakage channel is switched OFF, and the I_OFF_ is decreased with the increasing N_S_. This is because large N_S_ can reduce the electron concentration flowing from the source region to the channel, thereby inhibiting the leakage current composed of electron current in the OFF-state. Meanwhile, Figure 4b shows rapid I_ON_ increase due to the increasing of ON-state BTBT rate.

The HJ-LTFET is robust to variations in channel doping concentration (N_C_) and drain doping concentration (N_D_). Changes in N_C_ and N_D_ have little effect on the transfer characteristic, I_ON_, and I_ON_/I_OFF_, as shown in Figure 5 and Figure 6. The increase of N_C_ and N_D_ will reduce the channel resistance, resulting in a slight increase in the I_ON_, as shown in Figure 5b and Figure 6b. However, when the drain doping increases to 1 × 10^18^ cm^−3^, Figure 6b shows that the I_OFF_ will increase sharply, resulting in a decrease in the I_ON_/I_OFF_. This is because the tunneling barrier between source and drain region will decrease with the increase of N_D_. When the barrier is lower than a certain value, SDT leakage current will be generated.

### 3.2. Effect of Geometric Dimensions on Device Performance

Figure 7 shows the change of the transfer characteristic, ON-state current, and OFF-state current of HJ-LTFET with different source thickness (*T*_S_). It is not difficult to find that, with the increasing *T*_S_, the I_ON_ increases continuously, and the variation of I_OFF_ can be ignored, which is maintained at around 6 × 10^−12^ μA/μm. This is because the increase of *T*_S_ promotes the generation of free electrons in the source region, thereby increasing the number of tunneling electrons from source to channel and ultimately increasing the current density in the ON-state.

Figure 8 shows the change of the transfer characteristic, ON-state current, and switching current ratio of HJ-LTFET with different channel thickness (*T*_C_). Since the increase of *T*_C_ increases the area of diagonal tunneling between source and drain, the I_ON_ increases with the increase of *T*_C_, as shown in Figure 8b. However, when the *T*_C_ is greater than 10 nm, it will also form an extremely serious tunneling current in the OFF-state, resulting in a decline in device switching performance. In general, it is necessary to compromise between high I_ON_ and low *I*_OFF_.

Figure 9 compares the effect of traditional low-doped spacer layer and modified HFO2 barrier layer with varying thickness between the source and drain on suppressing SDT leakage in OFF-state. It is not hard to find, in Figure 9a, that the SDT phenomenon (consisting of source-to-drain horizontal tunneling and source-to-drain diagonal tunneling) of the traditional structure in OFF-state is particularly serious. After the HfO2 barrier layer is used, the OFF-state electron tunneling on the horizontal direction of source and drain is completely suppressed, which shows that the performance of the modified structure has been greatly improved. For the diagonal tunneling between source and drain, the OFF-state leakage can be further reduced by increasing the thickness of the barrier layer, as shown in Figure 9b. This is because the increased barrier thickness weakens the drain voltage and increases the barrier height of SDT. At the same time, the I_ON_ marginally declines as the barrier layer’s thickness increases.

### 3.3. Effect of Metal Work-Function on Device Performance

Figure 10 shows the impact of multi-metal work function modulation on device performance. It can be seen from Figure 10a that the I_OFF_ is reduced by 10 orders of magnitude through raising the work function of the gate2 (WF_2_) to 4.85 eV. When the WF_2_ is improved further, the I_OFF_ changes slightly, and the I_ON_ will decrease gradually. Therefore, the WF_2_ can be appropriately improved by the work function modulation method to optimize the device structure.

In order to analyze the reasons for the above results, the distribution of energy band with or without work function modulation in OFF-state is shown in Figure 10b. The source-to-drain diagonal tunneling is the main cause of OFF-state leakage, and increasing *WF*_2_ appropriately can reduce the probability of such tunneling. This is because the energy band close to the drain is raised with the increase of *WF*_2_, and the elevation of the conduction band in the drain region increases the barrier of SDT. Therefore, by the application of multi-metal work function, the OFF-state tunneling phenomenon can be suppressed greatly.

### 3.4. Effect of Channel Length on Device Performance

Figure 11 shows the curve of subthreshold swing (SS) varying with channel length. It is clear to find that the SS changes weakly as the channel length decreases in the vertical and horizontal directions (V_ch_ and H_ch_). The SS is still lower than 60 mv/dec at Id = 0.1 μA, even if the channel length is reduced to 5 nm in both vertical and horizontal directions. This is because the L-shaped channel expands the length to a certain extent by making full use of the vertical space. Compared with conventional planar TFET, the short-channel effects is not prominent with the scaling of the device. Therefore, this structure has excellent scaling capability and device reliability.

### 3.5. Analog/RF Performance of the HJTFET

Figure 12a shows the transconductance curves of the HJTFET at Vds = 0.5 V. The transconductance (g_m_) can be obtained from the first derivative of the transfer characteristic curve [29], as shown in Equation (1):(1)$$g_{m}=dI_{ds}/dV_{gs}$$

As a result, the maximum transconductance of 947 μS/μm can be achieved at Vg = 0.5 V, as shown in Figure 12a. This is benefited from the high current gain contributed by InGaAs/GaAsSb heterojunction and L-shaped gate.

It is generally believed that the parasitic capacitance of devices is crucial to the frequency characteristics of integrated circuits, especially the gate capacitance (C_gg_). For proposed device structure, C_gg_ generally consists of gate-to-source capacitance (C_gs_) and gate-to-drain capacitance (C_gd_). Therefore, the characteristics of C_gg_, C_gs_, and C_gd_ are of great significance for evaluating the frequency characteristics and analog application ability of devices. As shown in Figure 12b, the C_gd_ of the HJTFET under 0.5 V gate voltage is 0.15 fF/μm at Vd = 0.5 V, which is much smaller than that of the C_gs_ (2.0 fF/μm at Vd = 0.5 V). Thus, the C_gg_ of the HJTFET is mainly determined by C_gs_, unlike the general heterojunction devices. This is closely related to the bias voltage applied on the gate and drain.

In addition to parasitic capacitance, cut-off frequency (f_T_) and gain bandwidth product (GBP) are also significant performance indicators to evaluate the frequency characteristics of devices. The fT depends on the ratio of transconductance to total capacitance [37], as shown in Equation (2). If a specific DC gain of 10 is assumed, the GBP can be expressed by Equation (3) [37,38].
(2)$$f_{T}=g_{m}/2\pi (C_{gs}+C_{gd})=g_{m}/2\pi C_{gg}$$
(3)$$GBP=g_{m}/2\pi 10C_{gs}$$

Figure 12c shows the characteristic curves of f_T_ and GBW versus Vg at V_d_ = 0.5 V. Benefiting from structural advantages, such as L-shaped channel and source, the HJTFET obtains the most outstanding frequency characteristics compared with others [21,22,23,24,25,26,27,28,29,30,31,32]. As shown in Figure 12b, the f_T_ and GBP of the HJTFET reached the maximum values of 68.3 GHz and 7.3 GHz under the condition of V_d_ = 0.5 V, respectively.

By comparison with similar HJ-TFET, the superior DC and high-frequency characteristics of the proposed structure are highlighted, as shown in Table 1.

## 4. Conclusions

In this paper, a novel InGaAs/GaAsSb HJ-LTFET device structure based on the line tunneling mechanism is proposed. To reduce the leakage current in the OFF-state, the insertion of a HfO2 barrier layer separates the source region from the drain region to prevent the source-to-drain horizontal tunneling leakage. In addition, by adjusting the electrode work function, the gate metal work function above the drain region is improved to suppress the source drain diagonal tunneling. Meanwhile, the InGaAs/GaAsSb heterojunction of quasi-fault gap energy band alignment as well as large overlap area with an L-shaped between source region and channel layer greatly improve the I_ON_ of the device. Through TCAD simulation, the HJ-LTFET with a large I_ON_ of 213 μA/μm, a steep average SS of 8.9mV/dec, and a maximum fT and GBP of 68.3 GHz and 7.3 GH, respectively, can be obtained. Therefore, the device structure is expected to be widely used in ultra-low power circuits in the future.

## Figures and Tables

**Figure 1 micromachines-13-01474-f001:**
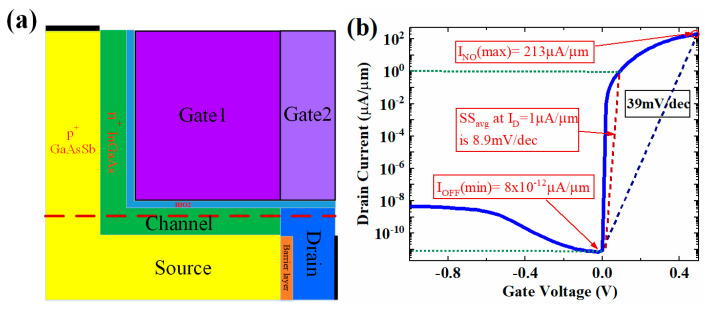
(**a**) Structure of the proposed InGaAs/GaAsSb HJ-LTFET; (**b**) the transfer characteristic of HJ-LTFET.

**Figure 2 micromachines-13-01474-f002:**
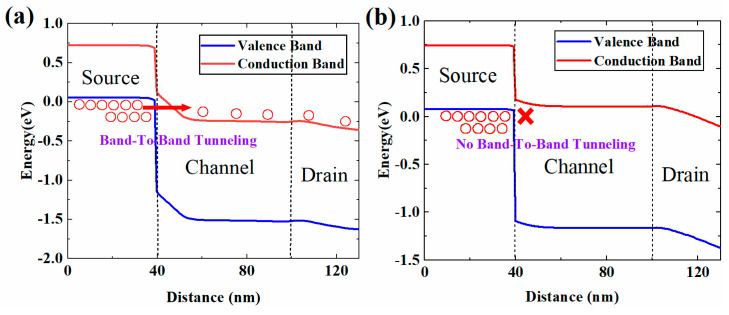
Band diagram in the (**a**) ON-state and (**b**) OFF-state.

**Figure 3 micromachines-13-01474-f003:**
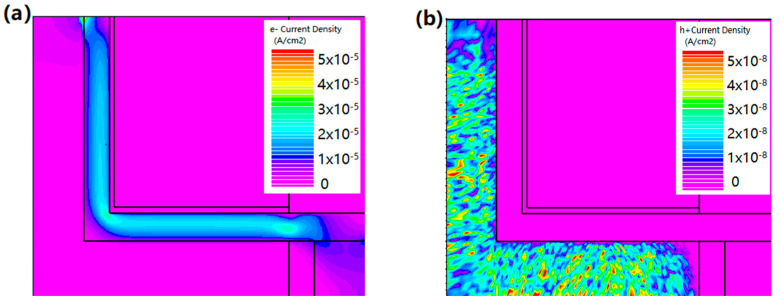
Simulated diagrams of (**a**) OFF-state electron current density, (**b**) OFF-state hole current density, (**c**) ON-state electron current density, and (**d**) ON-state hole current density at V_DS_ = 0.5 V.

**Figure 4 micromachines-13-01474-f004:**
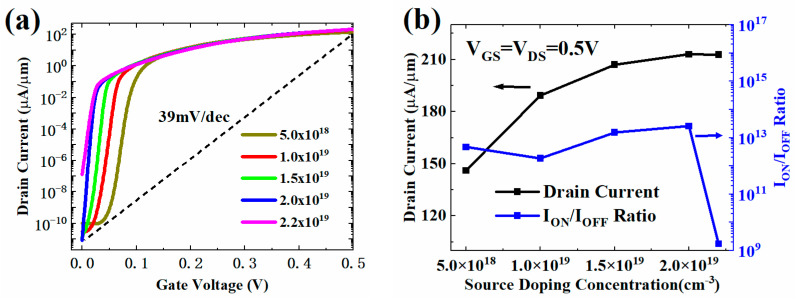
(**a**) Transfer characteristic and (**b**) I_ON_ and I_ON_/I_OFF_ with different N_S_ values.

**Figure 5 micromachines-13-01474-f005:**
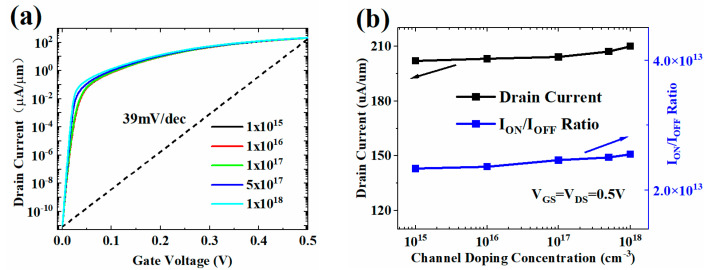
(**a**) Transfer characteristic and (**b**) I_ON_ and I_ON_/I_OFF_ with different N_C_ values.

**Figure 6 micromachines-13-01474-f006:**
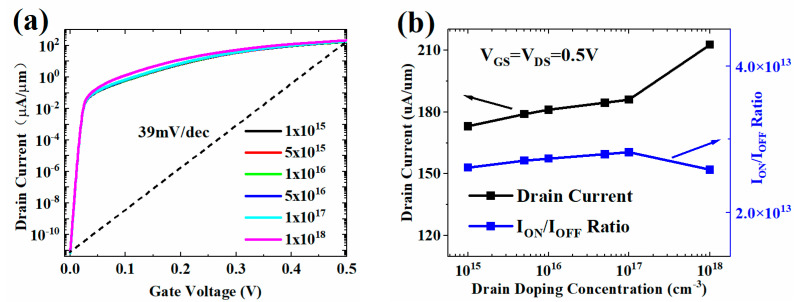
(**a**) Transfer characteristic and (**b**) I_ON_ and I_ON_/I_OFF_ with different N_D_ values.

**Figure 7 micromachines-13-01474-f007:**
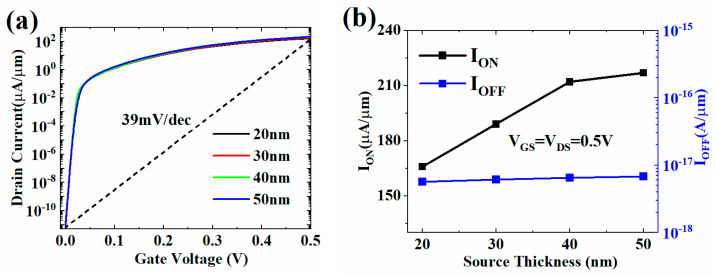
(**a**) Transfer characteristic and (**b**) *I*_ON_ and *I*_ON_/*I*_OFF_ with different *T*_S_ values.

**Figure 8 micromachines-13-01474-f008:**
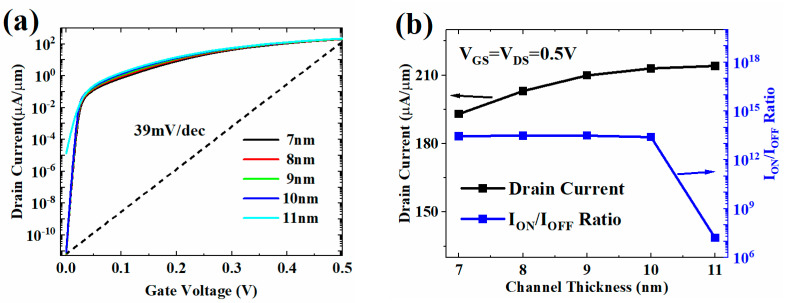
(**a**) Transfer characteristic and (**b**) *I*_ON_ and *I*_ON_/*I*_OFF_ with different T_C_ values.

**Figure 9 micromachines-13-01474-f009:**
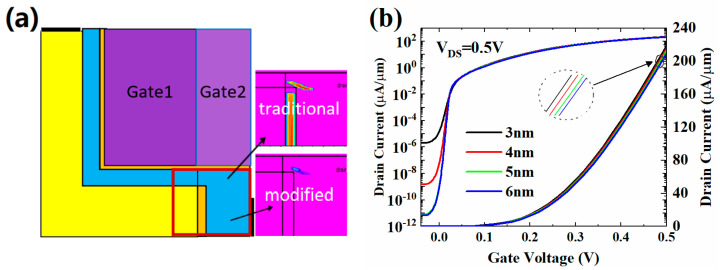
(**a**) OFF-state tunneling rate distribution with traditional low-doped spacer layer or modified HFO2barrier layer and (**b**) transfer characteristic curves with different thickness of HfO2 barrier layer.

**Figure 10 micromachines-13-01474-f010:**
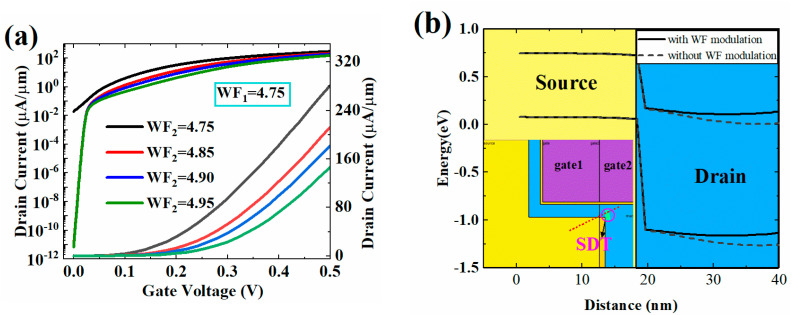
(**a**) Transfer characteristic and (**b**) band diagram with work-function modulation.

**Figure 11 micromachines-13-01474-f011:**
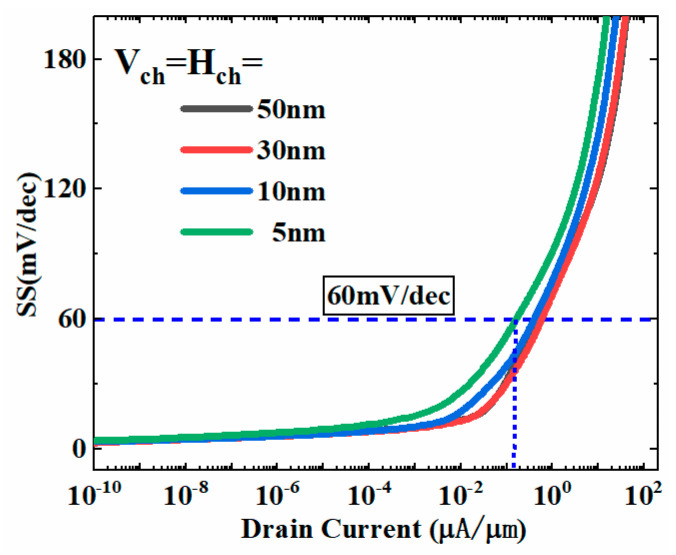
Subthreshold swing curve varying with channel length.

**Figure 12 micromachines-13-01474-f012:**
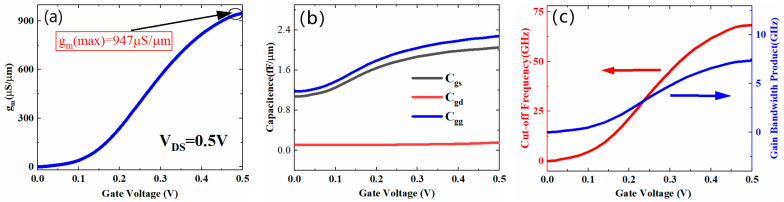
(**a**) Transconductance curves, (**b**) parasitic capacitance curves, and (**c**) f_T_ and GBP curves of the HJTFET.

**Table 1 micromachines-13-01474-t001:** Comparison with key parameters of similar device structures.

	InGaAs/InP HJ-TTFET [28]	InGaAs/InP HJ-LTFET [14,28]	Proposed InGaAs/GaAsSb HJ-LTFET
Channel length *Lch* (nm)	20	20	20
Drain voltage *Vds* (V)	0.5	0.5	0.5
Gate voltage *Vgs* (V)	0.5	0.5	0.5
Source doping *Ns* (cm^−3^)	5 × 10^19^	5 × 10^19^	2 × 10^19^
Drain doping *Nd* (cm^−3^)	5 × 10^18^	5 × 10^18^	1 × 10^18^
*Ion* (μA/μm)	163	85	196
*Ion/Ioff*	5 × 10^8^	4 × 10^7^	2 × 10^13^
*gm* (μS/μm)	500	140	950
*Cgg* (fF/μm)	1.7	1.0	2.2
*fT* (GHz)	46.8	22.3	68.3

## Data Availability

Data are contained within the article.

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
