# Peer review of "A Novel L-Gate InGaAs/GaAsSb TFET with Improved Performance and Suppressed Ambipolar Effect"

_micromachines, 2022, doi:10.3390/mi13091474_

Round 1
Reviewer 1 Report
In this paper, the authors tried to analyze a InGaAs/GaAsSb heterojunction TFET that is based on an L-shaped gate (HJ-LTFET). In this TFET, an HfO2 barrier layer is inserted between source and drain to suppress the horizontal tunneling leakage current. The presented structure has been studied by TCAD simulation. Different design parameters are investigated in order to optimize the device operation. After optimization, the HJ-LTFET achieves ION of 213 μA/μm, an average SS of 8.9 mV/dec and a suppressed IOFF of 10-20 μA/μm. The paper presents some new results; however, major revision is needed before resubmission. Here are some comments:
1. The introduction should be rewritten as it lacks a lot of information. A thorough survey is needed to indicate the difference between the current work and other similar studies. For instance, some of the authors of this paper published a similar work regarding the U-shaped InGaAs/GaAsSb heterojunction TFET [REF 23]. Authors are required to discuss the details of their previous work and give comprehensive discussion about the novelty of their study.
2. Before going to the simulation of the proposed structure, it is necessary to calibrate the materials system used in the device versus an experimental structure. Nonlocal band to band tunneling parameters used in simulation should be given.
3. It is important to study the analog performance parameters in parallel to the static parameters as sometimes increasing ION may lead to capacitance rise which limits the device usability in circuit applications. So, authors are required to extend their work to get the overall capacitance, transconductance, and/or cutoff frequency to give more insight about their device.
4. Choosing an HfO2 layer with 2 nm thickness may lead to gate leakage current as the EOT is very low (<0.35 nm). Also, this low EOT is difficult to be fabricated. Authors should discuss these issues.
5. It is necessary to compare the key parameters of the proposed structure versus the state of art structures in the literature. A table at the end of the manuscript is required to list the optimized parameters of the HJ-LTFET and those of other similar heterostructures.
6. Some references are not written in an appropriate way. Please, review all references.
Author Response
Dear reviewer, thank you for your time and effort on reviewing this manuscript!
Please see the attachment.

Reviewer 2 Report
The comments and suggestions are given below for the paper titled: A novel L-Gate InGaAs/GaAsSb TFET with improved performance and suppressed ambipolar effect
In the presented paper, A heterojunction tunneling field effect transistor with an L-shaped gate (HJ-LTFET) which is very applicable to operate at low voltage is proposed and studied by TCAD tools in this paper. InGaAs/GaAsSb heterojunction is applied in HJ-LTFET to enhance the ON-state current (ION). The proposed design exhibits high performances.
The investigated topic is interesting and original. The paper is well written and free from scientific mistakes. However, some points should be highlighted by the authors as given below:
- Some minor writing typos should be corrected.
- Fig.1a is not clear; I suggest adding another one by showing the different regions of the proposed structure (drain/channel/source and the oxide/gate).
- The introduction section should be improved by including some new sentences and updated references regarding the design and optimization of HJ-TFET. In this context, I suggest referring to: https://doi.org/10.1007/s12633-019-00190-w;https://doi.org/10.3762/bjnano.9.177; https://doi.org/10.1088/2053-1591/aac756.
- The scaling capability and device reliability should be discussed. In this context, I suggest adding a new Figure regarding the impact of the channel length on the subthreshold swing SS parameter. I suggest referring to: https://doi.org/10.1007/s10854-007-9531-y;
https://doi.org/10.1016/j.microrel.2010.10.002.
- The obtained results should be compared with other ones, recently published, in order to demonstrate the originality of this structure.
In summary, the achieved work showcases significant contributions. Thus, I would recommend with Major revision this manuscript for the possible publication in the Journal.
Author Response

(The authors gave the same response as above.)

Round 2
Reviewer 1 Report
In their revised manuscript, the author managed to address the raised issues. Some minor modifications are needed before the acceptance of this work. Here are some comments:
1. Some symbols should be revised like HfO2 (to be HfO2), fT (to be fT), Vd (to be VDS), ABTBT= (to be ABTBT =) and so on.
2. The units of Ns should be addressed in the caption of Fig. 4.
3. The units of NC should be addressed in the caption of Fig. 5.
4. The units of ND should be addressed in the caption of Fig. 6.
5. The symbols of capacitances should be in italic (Cgd to be corrected to Cgd and so on).
6. All symbols in Table 1 should be revised (for instance ION instead of Ion, fT instead of fT and so on).